# Differential Gene Expression Pattern of Importin β3 and NS5 in C6/36 Cells Acutely and Persistently Infected with Dengue Virus 2

**DOI:** 10.3390/pathogens12020191

**Published:** 2023-01-27

**Authors:** María Leticia Ávila-Ramírez, Ana Laura Reyes-Reyes, Rodolfo Gamaliel Avila-Bonilla, Mariana Salas-Benito, Doris Cerecedo, María Esther Ramírez-Moreno, María Elena Villagrán-Herrera, Ricardo Francisco Mercado-Curiel, Juan Santiago Salas-Benito

**Affiliations:** 1Doctorado en Ciencias en Biotecnología, Escuela Nacional de Medicina y Homeopatía, Instituto Politécnico Nacional, Mexico City 07320, Mexico; 2Campo Experimental Rosario Izapa, Instituto Nacional de Investigaciones Forestales, Agrícolas y Pecuaria, Tuxtla Chico, Chis 30878, Mexico; 3Institute of Immunology and Infection Research, School of Biological Sciences, University of Edinburgh, Edinburgh EH9 3JT, UK; 4Maestría en Ciencias en Biomedicina Molecular, Escuela Nacional de Medicina y Homeopatía, Instituto Politécnico Nacional, Mexico City 07320, Mexico; 5Facultad de Medicina, Universidad Autónoma de Querétaro, Santiago de Querétaro 76176, Mexico

**Keywords:** dengue virus, viral persistence, importin, nuclear transport

## Abstract

The establishment of persistent dengue virus infection within the cells of the mosquito vector is an essential requirement for viral transmission to a new human host. The mechanisms involved in the establishment and maintenance of persistent infection are not well understood, but it has been suggested that both viral and cellular factors might play an important role. In the present work, we evaluated differential gene expression in *Aedes albopictus* cells acutely (C6/36-HT) and persistently infected (C6-L) with Dengue virus 2 by cDNA-AFLP. We observed that importin β3 was upregulated in noninfected cells compared with C6-L cells. Using RT-qPCR and plaque assays, we observed that Dengue virus levels in C6-L cells essentially do not vary over time, and peak viral titers in acutely infected cells are observed at 72 and 120 h postinfection. The expression level of importin β3 was higher in acutely infected cells than in persistently infected cells; this correlates with higher levels of NS5 in the nucleus of the cell. The differential pattern of importin β3 expression between acute and persistent infection with Dengue virus 2 could be a mechanism to maintain viral infection over time, reducing the antiviral response of the cell and the viral replicative rate.

## 1. Introduction

Dengue (DEN) is one of the most rapidly spreading mosquito-borne viral diseases worldwide and has emerged as a public health problem with significant economic, political, and social impacts. The four dengue virus (DENV) serotypes (DENV-1, -2, -3, and -4) are transmitted to humans through the bites of infected mosquitoes of the genus *Aedes* (*Ae. aegypti* and *Ae. albopictus* species); each of these serotypes can cause a disease ranging from a self-limited febrile illness called dengue fever (DF) to dengue hemorrhagic fever (DHF) and dengue shock syndrome (DSS) [1].

DENV is an enveloped single positive-stranded RNA virus belonging to the Flaviviridae family. The DENV structure consists of a 50 nm-diameter particle with a lipid bilayer envelope with two associated structural proteins: the membrane (M) and the envelope (E). The envelope encloses the capsid (C) protein that surrounds the viral genome. After cell entry, the viral RNA is released into the cytoplasm and translated into a single polyprotein that is co and post-translationally processed by cellular and virus-derived proteases into three structural proteins, C, prM, and E, and seven nonstructural (NS) proteins, NS1, NS2A, NS2B, NS3, NS4A, NS4B, and NS5 [2].

Infection in humans occurs during the blood meals of infected female mosquitoes when virions are released along with the saliva. After some days of the intrinsic incubation period, individuals present different forms of febrile illness, as mentioned above. In mosquitoes, after ingestion of a DENV-infected blood meal from humans, the first virus targets are the anterior and posterior midgut epithelial cells [3]. The virus then escapes from the midgut basal lamina into the hemocoel and infects peripheral organs, including salivary glands. Finally, the virus is secreted into the saliva, and another transmission cycle takes place [4]. During this process, the mosquito vector acquires a life-long or persistent infection without apparent pathological or deleterious fitness effects. Persistent infections facilitate viral transmission to human hosts [5].

Long-term viral infections in cell lines provide an in vitro model to study chronic diseases. In the case of flavivirus, chronic infection is easily established in cell lines of mosquito origin. The mechanisms involved in the establishment and maintenance of persistent infection are not well understood, but it has been suggested that both viral and cellular factors might play an important role [4,5].

In addressing the viral and cellular factors that might be involved in the establishment and maintenance of DENV persistent infection, we generated an *Ae. albopictus* C6/36-HT cell line persistently infected with DENV-2 known as C6-L. This cell line displayed oscillating virus titres during a 32-week period, and then the virus titre decreased progressively until no virus was detected in culture media after 42 weeks by plaque assay in BHK-21 cells. Moreover, cytopathic effects (CPEs) were clearly observed during the first seven weeks, but they gradually diminished after that and were completely absent by week 30. On the other hand, the viral genome and some viral proteins [6], and viral particles with restricted replication capacity in mammalian cells [7] could be detected.

In the present research work, using this experimental model, we proposed and tested three related hypotheses. The first hypothesis to be tested is that there are genes potentially related to the DENV replicative cycle that are differentially expressed in noninfected cells when compared with persistently infected C6-L cells. This is relevant as it represents the identification of a likely genetic determinant of the mosquito’s intrinsic ability to support viral replication. The second hypothesis to be tested is that the kinetics of dengue virus replication over time are different in acute and persistently infected cells. In this way, it would be possible to have different models of viral infection that would allow the search for the respective cellular and molecular determinants. The third hypothesis to be tested is that the expression level of that previously identified as regulated upon DENV-2 infection importin β3 gene is also differentially expressed in mosquito cells acutely and persistently infected with DENV-2 over time, which could correlate with higher levels of NS5 in the nucleus of the cell. To establish a successful infection, DENV-2 requires an appropriated nuclear localization of NS5 protein, which is an active process that might require the participation of importin β3.

In this manuscript, we report the testing of these three hypotheses and present the results in the context of vector mosquito cell-DENV interactions during the early stages of infection.

## 2. Materials and Methods

### 2.1. Mosquito Cells

C6/36-HT cells from *Ae. albopictus* mosquitoes [8] adapted to grow at 35 °C [9], and C6-L55 cells (C6/36-HT cells persistently infected with DENV-2 for 55 weeks) were grown under the same conditions as previously reported [6]. BHK-21 cells (RRID:CVCL_1914, https://web.expasy.org/cellosaurus/CVCL_1915, accessed on 12 December 2022) were grown on plastic Petri dishes with the same culture medium for C6/36 cells at 37 °C in a humid incubator with 5% CO_2_ (Lab-Line 490, Barnstead, NH, USA).

### 2.2. Dengue Virus

DENV-2 New Guinea C strain (NGC) donated by the Instituto Nacional de Diagnóstico y Referencia Epidemiológicos, México (INDRE), was propagated in 2–3-day-old BALB/c mice, according to the protocol reported previously [10]. The plaque assay was carried out in BHK-21 cells for DENV-2 titration following the steps previously described by Juarez-Martínez et al. [6].

### 2.3. Amplified Fragment Length Polymorphism (AFLP)

This technique was performed as previously reported [11,12] with some modifications. Total RNA from noninfected (C6/36-HT) or persistently infected (C6-L55) cells was purified using TRIzol™ reagent (Invitrogen, Waltham, MA, USA) according to the manufacturer’s instructions. The contaminant DNA was removed with RQ1 RNase-Free DNase (Promega, Madison, WI, USA). Then, the mRNA was purified using the MicroPoly(A) Purist Procedure kit (Invitrogen, Waltham, MA, USA). This mRNA was used to synthesize cDNA with a SuperScript™ Double-Stranded cDNA Synthesis kit (Invitrogen, Waltham, MA, USA). The double-stranded DNA was subjected to digestion sequentially with EcoRI and MseI (Biolabs, Boston, MA, USA) restriction endonucleases, as previously reported [11]. The EcoRI (5 pmol) and MseI (50 pmol) adaptors (see Appendix A) were ligated to the digested cDNA using 5 U of T4 DNA ligase (Biolabs, Boston, MA, USA) in a final volume of 40 µL for 48 h at 37 °C. Then, the pre-selective PCR was performed by adding 10 × buffer (2.5 µL), MgCl_2_ (1.5 µL), 10 mM dNTP mix (2.5 μL, Biolabs, Boston, MA, USA), 10 µMol of both EcoRI and MseI pre-selective primers (2.5 µL, see Appendix A), 5 µL of cDNA (previously ligated to the adaptors and diluted 1:10), 0.5 µL of Taq DNA polymerase (5 U/μL, Promega, Madison, WI, USA) and nuclease-free water to a final volume of 50 μL. The reaction was performed in a Thermal Cycler (Eppendorf Mastercycle, Hamburg, Germany) using the following conditions: 1 cycle for 5 min at 94 °C, 30 cycles consisting of 1 min at 94 °C for denaturalization, 1 min at 56 °C for annealing, and 1 min at 72 °C for extension. A final cycle at 72 °C for 10 min was added. The reaction was diluted 1:10 and used for selective PCR. For that, 48 combinations of selective primers were used in the PCRs as follows: ES1/MS1, ES1/MS3, ES1/MS4, ES1/MS5, ES1/MS6, ES1/MS8, ES1/MS10, ES1/MS11, ES2/MS1, ES2/MS3, ES2/MS4, ES2/MS5, ES2/MS7, ES2/MS10, ES2/MS11, ES2/MS13, ES3/MS1, ES3/MS2, ES3/MS3, ES3/MS4, ES3/MS6, ES3/MS7, ES3/MS8, ES3/MS9, ES3/MS10, ES3/MS11, ES3/MS12, ES3/MS13, ES4/MS1, ES4/MS2, ES4/MS4, ES4/MS5, ES4/MS6, ES4/MS7, ES4/MS10, ES4/MS11, ES4/MS12, ES4/MS13, ES5/MS10, ES6. The reactions were performed by adding 10 × buffer (5 µL), MgCl_2_ (5 µL), 10 mM dNTP mix (2 μL, Biolabs, Boston, MA, USA), 10 µmol of both EcoRI and MseI selective primers (2 µL, see Appendix A), 3 µL of DNA (from the pre-selective PCR and diluted 1:10), 0.5 µL of Taq DNA polymerase (5 U/μL, Promega, Madison, WI, USA) and nuclease-free water to a final volume of 50 μL. The reaction was performed in a Thermal Cycler (Eppendorf Mastercycle, Hamburg, Germany) using the following conditions: 1 cycle for 10 min at 94 °C, 5 cycles consisting of 1 min at 94 °C for denaturalization, 1 min at 65 °C for annealing, and 1 min at 72 °C for extension. Then, 5 cycles consisted of 1 min at 94 °C, 1 min at 60 °C, and 1 min at 72 °C. Then, 30 cycles consisted of 1 min at 94 °C, 1 min at 56 °C, and 1 min at 72 °C. A final cycle at 72 °C for 10 min was added. All reactions were carried out in triplicate and analyzed by electrophoresis in 12% polyacrylamide gels stained with ethidium bromide. The differential amplicons were excised from the gel and incubated in diffusion buffer (0.5 M ammonium acetate, 10 mM magnesium acetate, 1 mM EDTA pH 8.0, and 0.1% SDS) at 50 °C for 30 min at a ratio of 100–200 µL of buffer/100 mg of gel. Then, the amplicons were purified using the QIAquick Gel Extraction Kit (QIAGEN, Hilden, Germany) according to the manufacturer’s protocol. After purification, the amplicons were cloned into a pJet vector (Thermo Fisher Scientific, Waltham, MA, USA), transformed into DH5a *E. coli* cells, purified with a GeneJET™ Plasmid Miniprep Kit (Fermentas, Waltham, MA, USA), and sequenced in Unidad de Biología Molecular at Instituto de Fisiología Celular of Universidad Nacional Autónoma de México using ABI Prism 310 and ABI Prism 3100 systems by Applied Biosystems. The sequences were visualized using BioEdit 7.0 software and analyzed by nucleotide BLAST software and the GenBank database GenBank (RRID:SCR_002760) of the National Center for Biotechnology Information (NCBI) (http://www.ncbi.nlm.nih.gov/BLAST, accessed on 29 November 2011).

### 2.4. Bioinformatic Analysis of Importin from Ae. albopictus

The amino acid sequence of Importin β3 of *Ae. albopictus* (AALF002645-RA) obtained from VectoBase VectorBase Release 61 (RRID:SCR_005917, https://vectorbase.org/vectorbase/app/, accessed on 1 April 2022) was used for multiple sequence alignment with Clustal Omega Version 2019 (RRID:SCR_001591, https://www.ebi.ac.uk/Tools/msa/clustalo/, accessed on 7 April 2022) and for structural alignment with T-COFFEE software T-Coffee Version 3 (RRID:SCR_011818, http://tcoffee.crg.cat/, accessed on 10 April 2022) with different importin protein sequences from Diptera species: *Ae. aegypti* (AAEL010159-PA), *Culex quinquefasciatus (Cx quinquefasciatus*, CPIJ010329-RA), *Anopheles gambiae* (*An. gambiae*, AGAP003769-PA), and *Drosophila melanogaster (D. melanogaster*, Q9VN44_DROME); and humans (NP_002262.4). Clustal Omega phylogenetics results were visualized using iTOL Version 6 (RRID:SCR_018174, https://itol.embl.de/, accessed on 10 April 2022). Additionally, all proteins were modeled by SWISS-MODEL Version 2018(RRID:SCR_018123, https://swissmodel.expasy.org/, accessed on 15 April 2022), and structural analysis of importins was performed using Prosite software Version 2022, (RRID:SCR_003457, https://prosite.expasy.org/, accessed on 24 April 2022) to describe the structural organization. The motif sequences identified from different importins were aligned with Clustal Omega to define a possible association of function between the proteins.

### 2.5. DENV-2 Infection Time-Course Assays

C6/36-HT cells were seeded in 24-well plates (Corning, Corning, NY, USA). A total of 5 × 10^5^ cells per well were incubated overnight at 35 °C. Cell monolayers were infected with DENV-2 at a multiplicity of infection (MOI) of 0.1, 2, and 10 for 2 h at 37 °C with gentle shaking. Cells were washed extensively with phosphate-buffered saline (PBS) supplemented with 0.5% FBS and incubated further with fresh medium. At the indicated postinfection time points (12, 24, 36, 48, 72, and 120 h), culture media were used for virus titration by plaque assay in BHK-21 cells as previously described by Juarez-Martínez et al. [6], and infected cells were harvested for total RNA purification. Mock-infected C6/36-HT cells where an extract of noninfected suckling mouse brain was used as inoculum were included for each MOI as a control. Additionally, cell viability was evaluated by cell proliferation kit I (MTT) [3 (4,5-dimethylthiazol-2-yl)-2,5-diphenyltetrazolium bromide] (MTT) (Roche, Basel, Switzerland) according to the manufacturer’s instructions. Briefly, C6/36-HT cells were grown in 96 well tissue culture plates at a concentration of 2.5 × 10^4^ cells/well in a 200 µL culture medium under the same conditions mentioned above. At the indicated postinfection time points, cells were incubated with MTT solution for approximately 4 h. Then, a solubilization solution was added and incubated overnight in a humidified atmosphere, and optical density was determined between 550 and 600 nm.

Mock-infected C6-L55 cells were inoculated with noninfected suckling mouse brains for each respective MOI following the same procedure mentioned above. The C6-L55 cell monolayers were harvested at the same time as C6/36-HT cells for total RNA purification. The assay was carried out in triplicate. Isolated RNA was further processed for importin-β3 gene expression level and DENV-2 copy number determinations.

### 2.6. RNA Purification and Treatment

Total RNA was purified from mock-infected and DENV-2 acutely infected C6/36-HT cells and mock-infected C6-L55 cells. TRIzol^®^ Reagent (Invitrogen, Waltham, MA, USA) was used following the manufacturer’s instructions. Purity and concentration were verified by determining the 260/280 ratio and the absorbance at 260 nm in a BioTek Epoch Spectrophotometer System. To remove contaminating DNA from total RNA preparations, samples were treated with a TURBO DNA-free™ (Applied Biosystems, Waltham, MA, USA) kit following the manufacturer’s specifications.

Total RNA obtained in this way was used as follows for the verification of the presence of the viral genome in the cell cultures or the cDNA synthesis for determination of importinβ3 expression level and DENV-2 genome copy number.

### 2.7. cDNA Synthesis

Single-stranded cDNA was synthesized from total RNA using the ImProm-II Reverse Transcription System kit (Promega, Madison, WI, USA) according to the manufacturer’s recommendations. Briefly, 1.0 µg of total RNA was mixed with 0.5 µg of random primer and incubated for 5 min at 70 °C. The reaction mixture (H_2_O, 5 × Buffer, MgCl_2_, dNTPs, RNasin RNase inhibitor, enzyme RT) was added and incubated for 5 min at 25 °C, 60 min at 42 °C and finally 15 min at 70 °C.

### 2.8. Verification of the DENV-2 Genome

To verify the persistent infection in C6-L cells, total RNA from C6-L55 mock-infected and DENV-2 acutely infected C6/36-HT cells was analyzed by RT–PCR using a MyTaq™ One-Step RT–PCR kit (Bioline, Cincinnati, OH, USA) according to the manufacturer’s instructions. Specific primers DV2M15 and DV2M16 were used for the amplification of an NS4A-NS4B-NS5 region of the DENV-2 genome, as previously reported by Juárez-Martínez [6]. *Ae. aegypti* S7 rRNA was used as an endogenous control using specific primers, as previously reported by Reyes-Ruiz et al. [7]. Amplicons were analyzed by gel electrophoresis.

### 2.9. Quantification of the DENV-2 Genome

Total RNA extracted from DENV-2 acutely infected C6/36-HT and mock-infected C6-L55 cells from the time-course assays were analyzed by RT–qPCR to determine DENV-2 genome copy number according to the protocol reported by Avila-Bonilla et al. [13]. A standard curve from 10^8^–10^2^ copies of the DENV NS5 gene was generated with a recombinant plasmid, as previously described [13]. A relative standard curve qPCR assay was carried out in triplicate for cDNA samples and serial dilutions of the recombinant plasmid using TaqMan^®^ 2× Universal PCR Master Mix (Applied Biosystems, Waltham, MA, USA) according to the manufacturer’s recommendations. Briefly, the reaction mixture consisted of 10.0 µL of TaqMan^®^ 2× Universal PCR Master Mix, 1 µL TaqMan^®^ Gene expression assay (20×), and 9 µL of DNA template. S7 rRNA was used to normalize the DENV NS5 gene copy number. Control reactions without DNA template or with cDNA from mock-infected C6/36-HT cells were carried out. The TaqMan^®^ Gene expression assay (20×) was designed by Applied Biosystems. The amplification conditions were 50 °C for 2 min, 95 °C for 10 min, 40 cycles of 95 °C for 15 s, and 60 °C for 1 min. A Stratagene Mx3005P QPCR System (Agilent Technologies, Santa Clara, CA, USA) was used. The number of DENV-2 genome copies was calculated by default from MxPro qPCR software (Stratagene 2007. https://www.agilent.com/en/product/real-time-pcr-%28qpcr%29/real-time-pcr-%28qpcr%29-instruments/mx3000-mx3005p-real-time-pcr-system-software/mxpro-qpcr-software-232751, accessed on 1 January 2018, version 4.10.

### 2.10. Importinβ3 Expression Level

*Aedes* importin β3 expression was evaluated by RT–qPCR in mock-infected and DENV-2 acutely infected C6/36-HT cells and mock-infected C6-L55 cells from the time course assays. Specific gene primers and TaqMan probes are shown in Appendix A.4. The mRNA levels were normalized against *Aedes* S7 rRNA.

The Premix Ex Taq Probe qPCR kit (Takara RR390 L, Kusatsu, Japan) was used according to the manufacturer’s recommendations. Briefly, the reaction mixture consisted of 2 µL of cDNA, 10.0 µL of Premix Ex Taq (Probe qPCR), forward primer (10 µM), reverse primer (10 µM), 2 µL TaqMan probe, and 6 µL water. Control reactions without cDNA were carried out to confirm the absence of contamination in the qPCR. The qPCR conditions for importin β3 were 95 °C for 35 s, 45 cycles of 95 °C for 21 s, 54 °C for 35 s and 60 °C for 30 s, and final extension at 4 °C for 10 min. qPCR conditions for S7 rRNA were 95 °C for 35 s, 45 cycles of 95 °C for 21 s and 60 °C for 30 s, and final extension at 4 °C for 10 min. All assays were performed in triplicate. The StepOne™ Real-Time PCR System (Applied Biosystems™, Waltham, MA, USA) was used. The relative transcript levels were determined by the ΔΔCt method [14]. Data are presented as 2^−ΔΔCt^ ± S.D. (*p* < 0.05) determined by two-way ANOVA.

### 2.11. Western Blot Analysis

C6/36-HT cells acutely infected with DENV-2 at an MOI of 10, mock-infected C6/36-HT and mock-infected C6-L55 cell monolayers were suspended and lysed with RIPA lysis buffer (1.0% NP-40, 150 mM NaCl, 1.0% sodium deoxycholate, 0.1%, sodium dodecyl sulfate (SDS), 50 mM Tris) containing a protease inhibitor cocktail (Roche 11697498001, Basel, Switzerland) to obtain total protein extracts. The protein concentrations were determined by Pierce BCA protein assay (Thermo-Scientific, 23225, Waltham, MA, USA). The protein extract was mixed with loading buffer (Tris-HCl, β-mercaptoethanol, SDS, bromophenol blue, glycerol) and boiled for 5 min. A total protein extract (30 μg when possible) was separated by 10% SDS–PAGE (Mini-Protean system, Bio-Rad, Hercules, CA, USA) as described previously [6] and electrotransferred onto a PVDF membrane (Immobilon^®^, Millipore, Burlington, MA, USA) using a Trans-Blot^®^ semidry electrophoretic transfer cell (Bio-Rad, Hercules, CA, USA) according to the manufacturer’s recommendations in Bjerrum-Schafer-Nielsen transfer buffer (Bjerrum and Schafer-Nielsen, 1986) at 0.8 mA/cm^2^ from 1 to 2 h. Immunoblotting was carried out using a mouse anti-karyopherin β3 (A-2) monoclonal antibody (Santa Cruz Biotechnology, Inc., Dallas, TX, USA, sc-514122), a mouse anti-β-actin (C4) monoclonal antibody (Santa Cruz Biotechnology, Inc., Dallas, TX, USA, sc-47778, RRID:AB_626632) was used as a loading control, and a rabbit anti-NS5 polyclonal antibody (Invitrogen, Waltham, MA, USA, PA5-32200, RRID AB_2549673) was diluted 1:80, 1:10,000 and 1:4000 respectively and incubated overnight at 4 °C. The proteins were detected using a chemiluminescence reaction with the Super Signal West Pico Chemiluminescent Substrate kit (Thermo-Scientific, Waltham, MA, USA) according to the manufacturer´s recommendations. All experiments were performed in triplicate.

### 2.12. Immunofluorescence with Anti-Importin β3 and Anti-DENV NS5

C6/36-HT and C6-L55 cells were grown on coverslips placed in 24-well plates. The immunofluorescence assay was carried out following the protocol described previously [15]. Briefly, C6/36-HT cells acutely infected with DENV-2 at an MOI of 10 and mock-infected C6-L55 cells were fixed with 4% p-formaldehyde (Sigma–Aldrich, Saint Louis, MO, USA) in PBS at room temperature for 2 h, washed with PBS and then permeabilized with 0.2% PBT (5% BSA, 0.2% Triton X-100 in PBS), incubated overnight at 4 °C with 1:10 anti-karyopherin-β3 (mouse monoclonal antibody, Santa Cruz Biotechnology, Dallas, TX, USA, sc-514122) and 1:200 anti-DENV-2 NS5 (rabbit polyclonal antibody, Invitrogen, Waltham, MA, USA, PA5-32200), washed with 0.1% PBT (0.1% Triton X-100), incubated with donkey anti-rabbit Alexa Fluor 594 (Invitrogen, Waltham, MA, USA, A21207, RRID AB_141637) and donkey anti-mouse Alexa Fluor 488 and washed with 0.1% PBT. Nuclei were stained with DAPI (4-6-diamidino-2-phenylindole dihydrochloride) and washed with PBS. Cells were mounted onto glass slides with Vecta Shield (Vector Laboratories, Burlingame, CA, USA) and examined by confocal microscopy (LSM 710 confocal microscope, Carl Zeiss, Jena, Germany). Unlabeled samples, non-specific binding of secondary antibodies, negative control with noninfected C6/36-HT cells, and DENV-2 infected C6/36-HT cells were used as controls to perform fluorescent microscopy. Additionally, different dilutions of primary and secondary antibodies were evaluated.

### 2.13. Confocal Microscopy Analysis

Confocal images were obtained using a confocal laser scanning microscope (LSM 710, Carl Zeiss, Jena Germany). Confocal microscopy was set up for all the experiments as follows: samples were excited with lasers of wavelength 405 nm (DAPI), 488 nm (Alexa Fluor 488), and 594 nm (Alexa Fluor 594); laser power 2%, 3%, and 10%, respectively. Field areas of 67.48 × 67.48 μm, resolution of 512 × 512 pixels, and depth of 8 bits were examined using a Plan-Apochromat 63 ×/1.40 Oil DIC M27objective, zoom 2.0 ×, scan unidirectional. A collection of consecutive image series from different focal planes with a depth separation of 1 µm for each image was acquired by using the scan mode of the z-stack, obtaining 11–18 slices, comprising a total between 3–5 image series at different locations. All images were stored in Tagged Image File Format (TIFF). The measurements of fluorescence intensity were performed by using ImageJ software (NIH, RRID:SCR_003070, https://imagej.net/, accessed on 24 April 2022).

### 2.14. Statistical Analyses

All data were analyzed using GraphPad Software Prism version 6.07 (GraphPad, RRID:SCR_000306, http://graphpad.com/, accessed on 12 June 2015) with two-way ANOVA and multiple comparisons by Tukey´s test or Bonferroni´s test for statistical analysis. Statistical significance was defined as *p* < 0.05.

## 3. Results

### 3.1. Identification of Importin β3 as a Dengue Virus-Regulated Gene

The amplified fragment length polymorphism (AFLP) technique [11,16] has been used to genotype populations of *Ae. aegypti* mosquitoes successfully [17,18,19,20]. A variation of this technique, referred to as cDNA-AFLP, is an easy tool to evaluate genome-wide expression levels and does not require prior knowledge of gene coding sequences [21,22,23,24].

Using 48 combinations of selective primers for PCRs, we obtained different amplicons in 30 out of those combinations. One hundred thirty-four amplicons ranging from 100 to 900 bp were obtained, with 69 differentially expressed: 42 in noninfected cells and 27 in persistently infected cells. Representative gels using the ES4-MS6 and ES4-MS4 selective primers are shown in Figure 1.

The differential amplicons were purified from the gels and sequenced. The sequences were aligned using nucleotide BLAST software, and the results are shown in Appendix A. Among the genes identified were inosine-5-monophosphate dehydrogenase, a salivary secreted peptide; adenylate cyclase, ferritin, peritrophin, ethanolamine-phosphate cytidyltransferase, molybdopterin cofactor synthesis protein, and several ribosomal proteins (see Table 1).

Interestingly, using ES4-MS4 selective primers, an amplicon of approximately 150 bp was only present in noninfected cells (see Figure 1B), and its sequence showed 96% identity with importin 5 (importin β3) of *Ae. aegypti* (XM 001654246.2, see Appendix A) and 98 and 99% with importin 5 of *Ae. albopictus* (XM 019685965.2 and XM 019697578.2, respectively). Using the corresponding amino acid sequence found in *Ae albopictus*, we performed an alignment analysis using BLAST, Clustal Omega, and T-COFFEE software (see Appendix A). We found a high identity with importin β3 of *Ae aegypti* and *Cx. quinquefasciatus* but also with importins β3 from other insects, such as *An. gambiae* and *D. melanogaster*. The identity with human importin β3 was only 55%, but the structural similarity was 94% (see Table 2).

These results were in agreement with the phylogenetic analysis (see Appendix A). The N-terminal importin β motif, which is important for binding to the Ran protein, was present in both mosquitoes and humans (see Appendix A). Since importin α/β-dependent nuclear import plays an important role during DENV infection [38,39,40,41], we further analyzed the expression level of importin β3 in mosquito cells acutely and persistently infected with DENV-2.

### 3.2. Models of Viral Infection

Acute infection was performed at different MOIs (0.1, 2, and 10) for 12, 24, 36, 48, 72, and 120 h as described in the Materials and Methods. The supernatant was recovered to determine the viral titer by plaque assay in BHK-21 cells, and total RNA was purified from the cells to evaluate the viral genome copy number.

For persistent infection, we used our previously established model of C6-L cells [6]. Since cells are already infected with DENV-2, they were only seeded in multiwell plates and treated with brain extract from noninfected newborn mice to mimic the process performed in acute-infected cells. Total RNA was analyzed at 12, 24, 36, 48, 72, and 120 h after mock infection. Since C6-L cells release viral particles with a restricted infection capacity [7], we could not carry out a plaque assay in BHK-21 cells, and the infection was evaluated by RT–qPCR.

The viral titer was determined in a DENV-2 time-course infection in acutely infected C6/36-HT cells at MOIs of 0.1, 2, and 10. In all cases, the first lytic plaques were detected at 24 h of infection (1.3 × 10^2^ PFU/mL at an MOI of 0.1, 1.4 × 10^3^ PFU/mL at an MOI of 2, and 6.6 × 10^3^ PFU/mL at an MOI of 10), and a gradual increase in viral titers was observed from 24 to 48 h without important changes between 24 and 36 h. The viral titers were the highest at 48 h of infection when MOIs of 2 and 10 were used (2.9 × 10^4^ PFU/mL and 5.0 × 10^4^ PFU/mL, respectively) and until 120 h at an MOI of 0.1 (3.5 × 10^4^ PFU/mL). The viral titers at 72 and 120 h, when MOIs of 2 and 10 were used, displayed significant oscillations, but they never decreased to the levels observed at 24 and 36 h (see Figure 2). Additionally, a similar experiment was performed in a 96-well tissue culture plate, and an MTT assay was performed. Any significant differences in cell viability were observed (see Appendix A).

The viral genome copy number was evaluated by RT–qPCR in both C6/36-HT cells acutely infected with DENV-2 and C6-L55 cells persistently infected with DENV-2 for 55 weeks. Since RT–qPCR is a more sensitive technique than plaque assays, the viral genome was detected as early as 12 h of infection in acutely C6/36-HT cells infected with DENV-2 at different MOIs (0.1, 2, and 10, 5.30 × 10^3^, 8.83 × 10^4^ and 1.51 × 10^5^ copies, respectively). Then, a gradual increase in the number of copies of the viral genome was detected from 12 to 48 h, as was observed in the plaque assay (6.6 × 10^6^ copies at an MOI of 0.1, 4.19 × 10^7^ copies at an MOI of 2 and 4.17 × 10^7^ copies at an MOI of 10 for 48 h, with the highest number of copies for 120 h of infection being 7.26 × 10^7^ at an MOI of 0.1, 1.20 × 10^8^ at an MOI of 2 and 6.72 × 10^7^ at an MOI of 10, see Figure 3). When an MOI of 0.1 was used, the increment extended to 72–120 h (5.74 × 10^7^ and 7.26 × 10^7^ copies to corresponding hours, see Figure 3A) but remained without important changes from 48 to 120 h when MOIs of 2 and 10 were used (the copy numbers for 48, 72 and 120 h at an MOI of 2 were 4.19 × 10^7^, 8.99 × 10^7^ and 1.20 × 10^8^, and at an MOI of 10 were 4.17 × 10^7^, 5.05 × 10^7^ and 6.72 × 10^7^ respectively, to the corresponding hours, see Figure 3B and C) according to the viral titers.

Since C6-L55 are already infected with DENV-2, we found higher levels of viral genome for 12 h at MOIs of 0.1, 2, and 10 (2.93 × 10^6^, 6.96 × 10^6^, and 5.59 × 10^6^ copies) than in C6/36-HT acutely infected at the same time point (5.30 × 10^3^, 8.83 × 10^4^ and 1.51 × 10^5^ copies to the corresponding MOI), and interestingly, they do not display important changes from 24 to 120 h. Statistically significant differences in viral genome copy number between DENV-2 acutely infected C6/36-HT, and C6-L cells were only detected at 72 and 120 h using MOIs of 0.1 and 2 and at 48, 72 and 120 h using an MOI of 10 (see Figure 3). Taken together, these results are consistent with the different viral replication behaviors between acute and persistent infections.

### 3.3. Importin β3 in Acute and Persistently DENV-2-Infected Cells

An acute and persistent model of infection in C6/36-HT mosquito cells was used to analyze the effect of DENV-2 on the expression of importin β3. We performed relative RT–qPCR in C6/36-HT cells infected with DENV-2 at different MOIs (0.1, 2, and 10) for 12, 24, 36, 48, 72, and 120 h. Mock-infected cells were used as a control.

The levels of importin β3 transcript remained unchanged at all different MOIs during the first hours of infection (12–36 h), then it increased at 48 h postinfection (1.66, 6.0, and 3.76 relative expressions at MOIs 0.1, 2, and 10, respectively) and remained at that higher level at 72 and 120 h of infection except when an MOI of 2 was used, where a decrease was observed (2.53 at 72 h and 2.68 at 120 h); however, it never reached its initial levels (Figure 4). Except for an MOI of 2 at 48 h, the level of importin β3 transcript was also associated with the viral inoculum at different time points: 1.67, 6.05, and 3.8 at 48 h; 1.89, 2.5, and 4.25 at 72 h; 1.9, 2.67, and 4.0 at 120 h; and 0.1, 2, and 10, respectively (Figure 4). Moreover, the relative mRNA level of importin β3 was also related to the DENV-2 genome copy number; at an MOI of 0.1, the highest viral genome copy number was detected at 72 and 120 h, and the highest level of importin β3 transcript was also detected (compare Figure 3A with 4A); at an MOI of 10, the highest viral genome copy number and the highest level of importin β3 transcript were detected at 48, 72 and 120 h (compare Figure 3C with Figure 4C).

Overall, in C6-L55 cells, no important changes in the levels of importin β3 mRNA were observed throughout the time points of infection (Figure 4), which was in agreement with the stable viral genome copy number (Figure 3). When we compared the importin β3 transcript levels in DENV-2 acutely infected C6/36-HT cells with the respective condition in C6-L55 cells, we observed a statistically significant higher level at 48, 72, and 120 h.

Additionally, we analyzed the expression of importin β3 by Western blot in total protein extracts from C6/36-HT cells infected with DENV-2 at an MOI of 10 and C6-L55 cells; actin was used for normalization (Figure 5A). The highest expression level of importin β3 was observed in acutely infected C6/36-HT cells at 72 h postinfection (Figure 5B), in agreement with the highest level of mRNA previously observed. The results with the same congruence were observed in C6-L55 cells, where lower expression levels of importin β3 were detected without statistically significant differences at all time points. Statistically significant differences among acutely and persistently infected C6/36-HT were observed at several time points of infection: 24, 36, and, once again, 72 h (Figure 5B).

### 3.4. Level of Expression and Localization of DENV NS5 in Acute and Persistently Infected Cells

C6/36-HT and C6-L55 cells were analyzed by confocal microscopy. No particular cellular localization pattern was observed for importin β3 in either model of infection (Figure 6), but the quantification of the corresponding fluorescent signal showed values consistent with what was observed in the quantification by Western blot (Figure 7A).

In immunofluorescence analysis by confocal microscopy, the NS5 protein was observed in the nucleus of C6-L55 cells as early as 12 h, and its level decreased over time until 48 h, when the corresponding signal could hardly be detected. This pattern was in agreement with that observed in the Western blot assay. In C6/36-HT cells acutely infected with DENV, the signal corresponding to the NS5 protein was barely evident in the cytoplasm at 24 h postinfection, and its levels increased considerably at 36 and 48 h, when it could be clearly observed in the nucleus of cells (Figure 6).

Taken together, these results suggest that DENV infection differentially modulates the expression of importin β3 in an acute infection than in a persistent infection.

## 4. Discussion

In the present study, we tested three hypotheses. The first hypothesis proposed that there are genes potentially related to the dengue virus replicative cycle that are differentially expressed in noninfected cells compared with persistently infected C6-L55 cells. Using a cDNA-AFLP approach, we found a higher expression level of importin β3 in noninfected C6/36-HT cells than in persistently infected C6/36-HT cells.

To establish a successful infection, the viruses required an appropriate subcellular localization of their proteins. The nucleus of eukaryotic cells is a double membrane organelle important for transcription and other levels of gene expression regulation. Several viruses, such as human immunodeficiency virus (HIV), influenza, respiratory syncytial virus (RSV), Rift valley fever virus, Venezuelan equine encephalitis virus, and human cytomegalovirus, among others [42], are required for cell components present in the nucleus and the migration of virus components into the nucleus to complete their replicative cycles successfully. Since the nuclear envelope is a barrier between the nucleus and the cytoplasm, the viruses need to overcome it to have access to this cellular compartment. This could be achieved by taking advantage of the temporary disassembly of the nuclear envelope during mitosis or using the nuclear-cytoplasm transport machinery [43].

The transport of molecules higher than 50 kDa through the nuclear pore complex is an active process that requires the participation of importins, represented mainly by importin α and β. The interaction of importin β with α enables the interaction of importin α with a highly basic amino acid sequence, named the nuclear localization signal (NLS), localized in cargo proteins. This, in turn, allows the translocation of a cargo protein into the nucleus through the nuclear pore. Additionally, some proteins are able to interact directly with importin β to obtain access to the nucleus [37].

Importin βs are a family of nucleocytoplasmic transport receptors that includes 10 proteins with nuclear import activities: β1, β2 (transportin-1), β2b (transportin-2), β3 (importin 5 or RanBP5), 4, 7, 8, 9, 11, and transportin SR (transportin 3 or importin 12); seven with nuclear export activities: exportin-1 (CRM1), exportin-2 (CAS/CSE1L), ex-portin-5, exportin-6, exportin-7, exportin-t and RanBP17; and two with bidirectional activities: importin-13 and exportin-4 [43]. Sequence analysis of importin β3 or importin 5 of *Aedes* mosquitoes revealed high homology with importins of other insects, such as *Culex*, *Anopheles*, and *Drosophila*, and high structural similarity with human importin β3. These proteins are composed of 19–20 HEAT repeats and have an N-terminal motif that binds to the Ran protein, an important conformational regulator of importins that determines the interaction with cargo molecules [44].

The second hypothesis proposed that the kinetics of dengue virus replication over time are different in acute and persistently infected cells. We found that the viral genome copy number in C6-L55 cells did not show a statistically significant difference at all time points, whereas in C6/36-HT cells, the number of copies of the viral genome increased over time, reaching its maximum levels at 72 and 120 h, which were also significantly higher than the levels observed in C6-L55 cells at those times.

The third hypothesis is that the expression level of that previously identified as regulated upon DENV-2 infection importin β3 gene is also differentially expressed in mosquito cells acutely and persistently infected with DENV-2 over time, which could correlate with higher levels of NS5 in the nucleus of the cell.

Overall, DENV acutely infected C6/36-HT cells expressed high importin β3 levels coincident with the highest viral titers and viral genome copy number. DENV persistently infected C6-L55 cells did not display important changes in viral genome copy number over the time of infection, and the expression of importin β3 was also stable.

Even though the flaviviruses replicate in the cytoplasm, several studies have demonstrated that many of their proteins migrate into the nucleus during infection and that the inhibition of cytoplasm-nucleus transport by compounds such as GW5074 [45], Ivermectin [46,47,48] and the synthetic retinoid N-(4-hydroxyphenyl) retinamide (4-HPR) [49,50] results in a reduction in the viral outcome. The analysis of the gene expression profile from the midgut of *Ae. aegypti* that differs in vector competence for DENV revealed that importin-β1 is upregulated upon infection in a susceptible strain [51], and the knockdown of Karyopherin 6 (KPNA6 or importin α7) inhibits Zika replication in Vero cells, which is reversed when the levels of this importin are restored [52]. Altogether, these results suggest that the cytoplasm-nucleus transport process is essential for the replicative cycle of flaviviruses.

NS5, an enzyme with methyltransferase and RNA-dependent polymerase (RdRP) activities, is one of the most studied DENV proteins that is translocated into the nucleus [53]. It has been localized to the nucleus of several mammals [38,39,41,47,48,54,55,56] and mosquito cells [54,55,57]. The nuclear localization of NS5 seems to be a common feature in several flaviviruses, such as WNV [58], JEV [59], and Zika [60,61]. Its translocation is mediated by the importin α/β complex [38,49,57,62] through its interaction with the NLS present in the RdRP domain [48,55,62,63] and the C-terminus of the protein [56]. The presence of NS5 in the nucleus is associated with virus outcome and a reduction in the production of IL-8, a cytokine with antiviral effects in the early stages of infection [38,39]. In HeLa cells infected with JEV, it has been reported that the NLS of NS5 is able to interact with KPNA3 (importin α4) and KPNA4 (importin α3), which play a role in the translocation of IRF3 and NFκB into the nucleus. Since these two transcription factors are important for the cellular antiviral response, this could be a viral mechanism to block it [59]. Additionally, the presence of NS5 in the nucleus has been implicated in changes in the expression of cellular genes to favor an environment suitable for viral replication [37,38,52,60,61].

There is no information regarding specifically importin β3 and flavivirus. However, it has been reported that it is able to interact directly with its cargos through their NLS [64] and its participation in the import of viral proteins such as Porcine circovirus type 2 [65], Human papillomavirus (HPV) [66,67], Influenza A virus (IAV), and HIV [68] has been well documented. Importin β3 is able to bind to the bipartite NLS located in the N-terminal segment of the PB1 subunit of RdRP of the IAV. This interaction allows the translocation into the nucleus of the dimer PB1-PA, where it binds to PB2 to form the complete and functional RdRP. Additionally, this interaction hides the 5′-vRNAp binding site of PB1-PA, keeping it in an inactive form during nuclear import. The silencing of importin β3 or mutations in the NLS of PB1 resulted in a reduction in viral RNA synthesis or viral particle production, respectively [69,70,71]. Moreover, there are reports that viral proteins can interact with more than one importin to ensure their access to the nucleus. For example, Rev, a retroviral transactivator protein, is translocated into the nucleus by the classic importin α/β pathway but is also capable of interacting directly with other importins such as transportin, importin 7, and importin 5 (β3), and this interaction is through the same arginine-rich NLS present in Rev [68]. The capsid protein L2 of HPV 16 is able to use several nuclear import pathways such as importin α2/β1, β2 or β3 [67]. In C6-L55 cells, low levels of importin β3, at least as low as in noninfected cells, could be relevant to maintain an antiviral status in the cell that is necessary for persistent viral infection. Those low levels of importin β3 in C6-L55 cells would be in agreement with the higher expression of NS5 at early time points when the cells are more actively proliferating, the virus is already established to replicate, and NS5 is already observed in the nucleus; then NS5 level decreases over time collaborating in maintaining the persistent viral infection. Most likely, the reduced and stable expression of importin β3 in C6-L55 cells affects the cellular environment to allow viral infection for long periods of time, unlike acutely infected C6/36-HT cells, where higher levels of importin β3 were observed at a time that a higher viral genome copy number was detected.

Unfortunately, there is no information regarding the role of importin β3 during other flaviviral infections, and the vast evidence of the participation of the importin β3 in the translocation of viral proteins comes from mammal cells, but the evidence presented here and with other viruses open the possibility that DENV NS5 could use importin β3 as an alternative molecule for nuclear import. However, with our results, we cannot prove the interaction of NS5 and importin β3, and this will require further investigation.

## 5. Conclusions

Taken together, these results clearly indicate differential expression of importin β3 between mosquito cells acutely and persistently infected with DENV-2 associated with the replicative rate of the virus, suggesting that importin β3 is one of the cellular factors associated with the maintenance of persistent infection. However, more studies will be necessary to define the exact role of cytoplasm-nuclear transport during DENV infection in mosquito cells.

## Figures and Tables

**Figure 1 pathogens-12-00191-f001:**
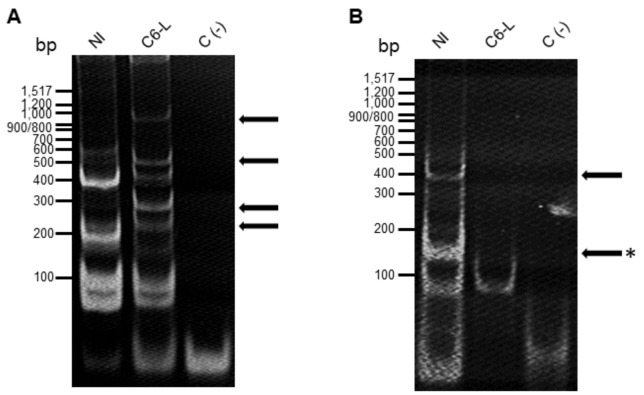
Representative of selective-PCR amplification. Purified mRNA from noninfected (NI) and persistently infected with DENV-2 (C6-L) C6/36 cells was used to synthesize cDNA and subjected to PCR amplification using the AFLP technique. The amplicons resulting from the selective PCR reactions were separated in a 12% polyacrylamide gel and stained with ethidium bromide. (**A**) Amplification using ES4-MS6. (**B**) Amplification using ES4-MS4 primers. The molecular markers are shown on the left side of the gels (100 bp Ladder New England Biolabs). Reactions without a cDNA template were included as a control (−). The arrows indicate differential amplicons. The asterisk indicates the amplicon corresponding to importin β3.

**Figure 2 pathogens-12-00191-f002:**
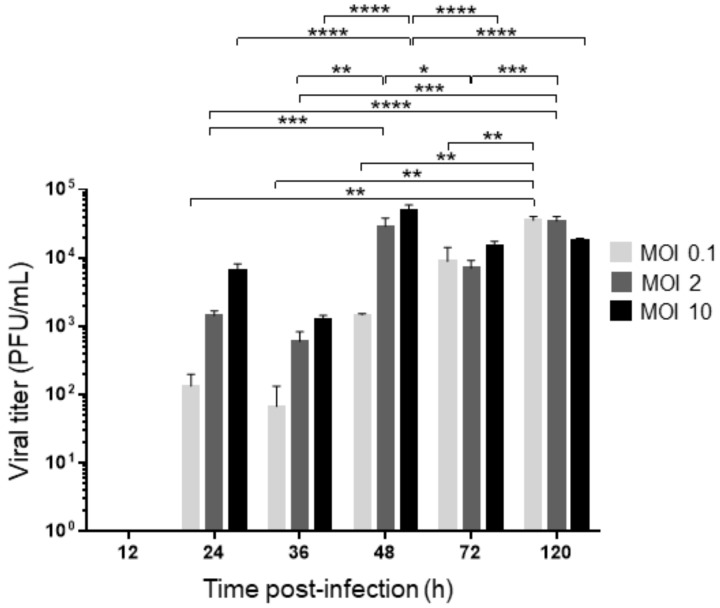
Viral titer in C6/36 cells at different MOI. Time course of viral titer in the supernatant of C6/36-HT cells infected with DENV-2 at an MOI 0.1, 2, and 10 at different times postinfection measured by plaque assay using BHK-21 cells. The experiment was carried out in triplicate. Results were expressed in PFU/mL and analyzed by two-way ANOVA and multiple comparison tests of Tukey. * *p* < 0.05, ** *p* < 0.01, *** *p* < 0.001, **** *p* < 0.0001.

**Figure 3 pathogens-12-00191-f003:**
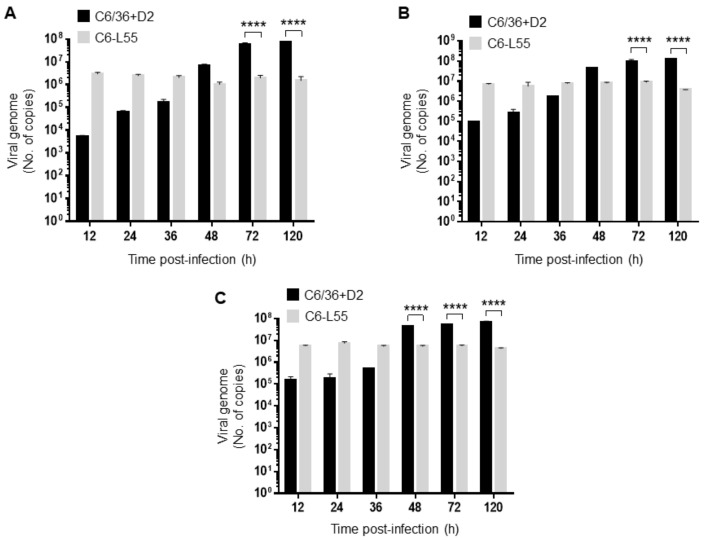
Determination of viral genome copy number. It was determined by RT-qPCR in C6/36 infected by DENV-2 (C6/36 + D2) at an MOI of 0.1 (**A**), 2 (**B**), and 10 (**C**) for 12, 24, 36, 48, 72, and 120 h. The same experiment was performed in C6/36 cells persistently infected with DENV-2 (C6-L55) using equivalent amounts of brain extract from noninfected mice as inoculum. The experiments were performed in triplicate. The results were analyzed by two-way ANOVA and multiple comparison tests of Bonferroni. **** *p* < 0.0001.

**Figure 4 pathogens-12-00191-f004:**
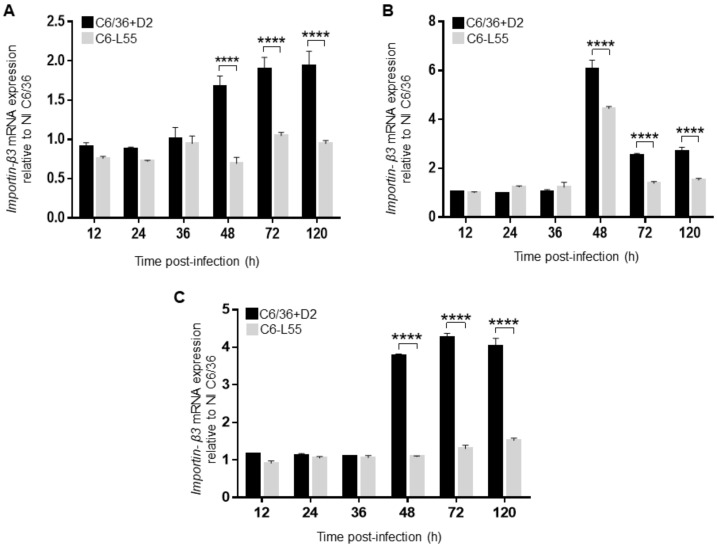
Expression of importin β3. It was determined by RT-qPCR in C6/36 infected by DENV-2 (C6/36 + D2) at an MOI of 0.1 (**A**), 2 (**B**), and 10 (**C**) for 12, 24, 36, 48, 72, and 120 h. The same experiment was performed in C6/36 cells persistently infected with DENV-2 (C6-L55) using equivalent amounts of brain extract from noninfected mice as inoculum. All experiments were performed in triplicate. The results were analyzed by two-way ANOVA and multiple comparison tests of Bonferroni. **** *p* < 0.0001.

**Figure 5 pathogens-12-00191-f005:**
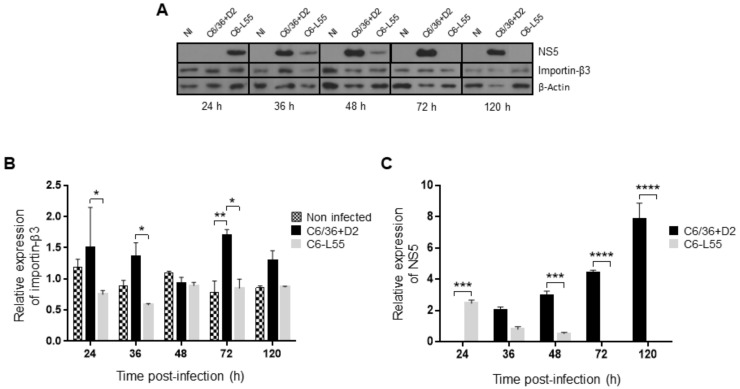
Expression of importin-β3 and NS5. (**A**) Importin-β3 and NS5 were determined by Western blot in DENV-2 infected C6/36 cells (C6/36 + D2) at an MOI of 10 at 24, 36, 48, 72, and 120 h. The same experiment was performed in DENV-2 persistently infected C6/36 cells (C6-L55) using equivalent amounts of brain extract from noninfected mice as inoculum. Mock-infected C6/36 cells (Noninfected, NI) were used as a control. Importin-β3 (**B**) and NS5 (**C**) protein bands were quantified using the ImageJ software densitometry analysis. Actin was used as a loading control to normalize all corresponding values. All experiments were performed in triplicate. Values shown are mean ± SEM from three independent experiments. The results were analyzed by two-way ANOVA and multiple comparison tests of Tukey (**B**) and Bonferroni (**C**). * *p* < 0.05, ** *p* < 0.01, *** *p* < 0.001, **** *p* < 0.0001.

**Figure 6 pathogens-12-00191-f006:**
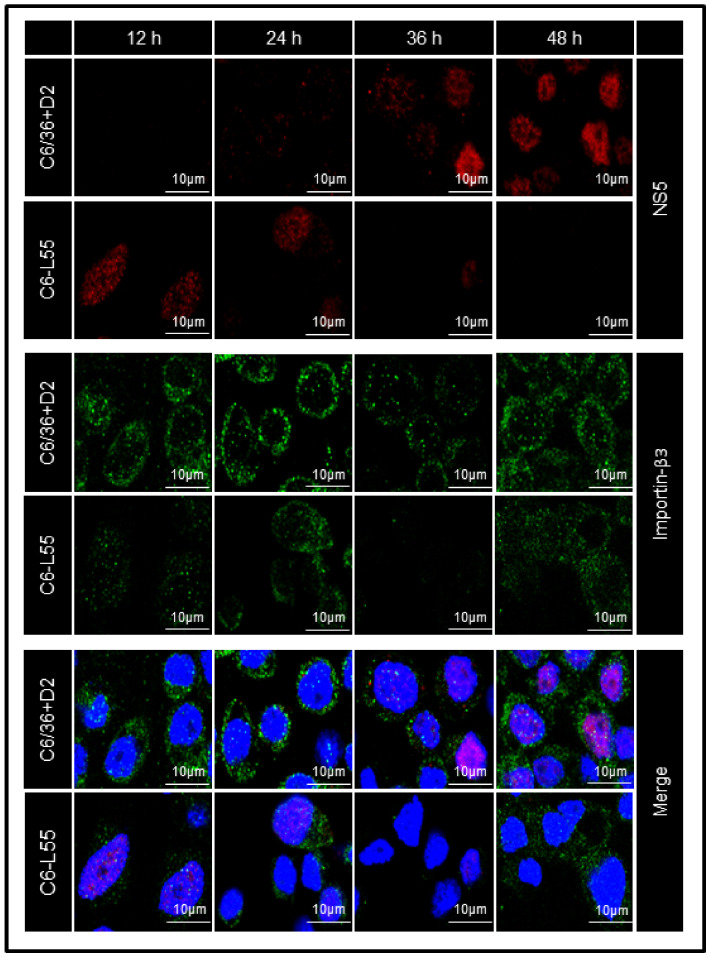
Subcellular distribution of Importin-β3 and NS5 proteins during the time course of DENV infection. DENV-2 infected C6/36 cells (C6/36 + D2) at an MOI of 10 at 12, 24, 36, and 48 h, DENV-2 persistently infected C6/36 cells (C6-L55) using equivalent amounts of brain extract from noninfected mice as inoculum at the same time were placed on glass coverslips and processed for double immunofluorescence labeling with specific antibodies anti-NS5 (red) and anti-karyopherin-β3 (Importinβ3, green) were followed by confocal microscopy. Nuclei were stained with DAPI (blue). Bar = 10 μm. Representative images are presented.

**Figure 7 pathogens-12-00191-f007:**
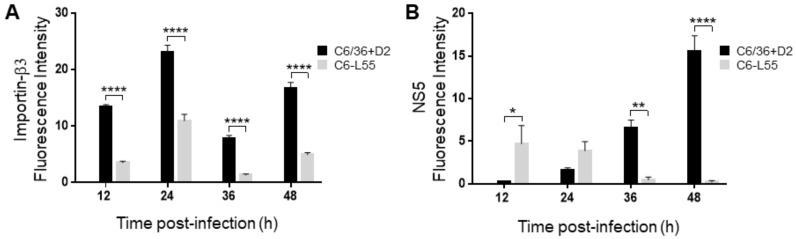
Immunofluorescence detection of importin β3 and DENV NS5. Immunofluorescence labeling with specific antibodies anti-NS5 and anti-karyopherin-β3 (Importin β3) was followed by confocal microscopy in DENV-2 infected C6/36 cells (C6/36 + D2) at an MOI of 10 at 24, 36 and 48 and hours. The same experiment was performed in DENV-2 persistently infected C6/36 cells (C6-L55) using equivalent amounts of brain extract from noninfected mice as inoculum. Quantification of Importinβ3 (**A**) and NS5 (**B**) is presented as the mean fluorescence intensity ± SEM. The measurements of fluorescence intensity were performed using the Image J software. Results were analyzed by two-way ANOVA and multiple comparison tests of Bonferroni. * *p* < 0.05, ** *p* < 0.01, **** *p* < 0.0001.

**Table 1 pathogens-12-00191-t001:** Functions of genes identified in the cDNA-AFLP technique.

Gene	Function	Reference
Inosine-5-monophosphate dehydrogenase	enzyme that participates in the biosynthesis of guanosine nucleotides but is also required for viral RNA synthesis in mosquitoes infected with ZIKV and DENV	[25,26,27]
Adenylate cyclase	enzyme that generates cyclic AMP from ATP	[28]
Ferritin	iron-binding protein that is expressed in the midgut, fat body, hemolymph, and ovaries, especially after a blood meal. This protein plays a role as a cytotoxic protector against the oxidative challenge during blood meal in mosquitoes	[29,30,31,32]
Peritrophin	component of the peritrofic matrix in the midgut of insects and crustaceans that stimulates the digestion of food and blocks the invasion of microorganisms	[33,34]
Ethanolamine-phosphate cytidyltransferase	one of the regulatory enzymes implied in the synthesis of ethanolamine-derived phospholipids, structural components of the cell membranes that regulate some cellular functions such as cell division, cell signaling, phagocytosis, and autophagy	[35]
Molybdopterin cofactor synthesis protein	Molybdenum cofactor–dependent enzymes play important roles in several biological processes in mammals, such as purine and sulfur catabolism. They participate in the oxidation of xenobiotics, and it has been associated with insecticide resistance in mosquitoes	[36]
Importin 5	Protein that participates in nuclear import	[37]

**Table 2 pathogens-12-00191-t002:** Bioinformatic analysis of importin β3.

AALF002645-RA Importin-5-like [*Ae. albopictus*]			T-COFFEE Structural Alignment	Clustal Omega	
*Organism*	Accession Number	Protein	Identity (%)	Identity (%)	E-Value
*Ae. aegypti*	AAEL010159-PA	importin beta-3	96%	99.09%	2.4 × 10^−58^
*Cx quinquefasciatus*	CPIJ010329-RA	Importin beta-3	96%	94.18%	1.4 × 10^−21^
*An. gambiae*	AGAP003769-PA	importin beta-3	95%	84.35%	7.7 × 10^−47^
*D. melanogaster*	Q9VN44_DROME	Karybeta3	95%	71.91%	2.9 × 10^−52^
*Homo sapiens*	NP_002262.4	importin-5	94%	55.45%	4.5 × 10^−59^

## Data Availability

All data generated or analyzed during this study are included in this article and its Appendix A. Further inquiries can be directed to the corresponding author.

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
