# Peer review of "Differential Gene Expression Pattern of Importin β3 and NS5 in C6/36 Cells Acutely and Persistently Infected with Dengue Virus 2"

_pathogens, 2023, doi:10.3390/pathogens12020191_

Round 1
Reviewer 1 Report
This is a very important work which would contribute significantly in understanding the mechanism of establishment and maintenance of persistent infection (DENV2) in mosquitoes. Although, we need more work which could define the exact role of cytoplasm-nuclear transport during DENV infection in mosquito cells. My comments are as follows. Line 432-434 "The highest expression level of importin β was observed in acutely infected C6/36-HT cells at 72 hours post-infection, in agreement with the highest level of mRNA previously observed." Comment - Importin-β bands for C6/36+D2 (Figure 5A, western blot) at 24 hrs and 36 hrs time points look more dense than that of of 72 hrs. Please explain. Why are loading control bands not consistent? Please explain. Please mention all the controls used in confocal microscopy (Figure 6) in the materials and method section.Author Response
Please, see the attachment

Reviewer 2 Report
This manuscript addressed understanding the differences between acute and persistent dengue virus infection by searching host molecules potentially related to the dengue virus replication in mosquito cells. Authors found different gene expression of importin β3 (referred to as “importin β” in this manuscript) which is the nucleocytoplasmic transport factor in mosquito cells between acute and persistent infection with Dengue virus 2. Second, it was reported that the kinetics of dengue virus replication were different in acute and persistently infected cells. Finally, the authors proposed that the expression of importin β correlated with the nuclear localization of the NS5 protein of the dengue virus, suggesting that importin β is an important factor in the maintenance of persistent infection with the dengue virus.
Comments: This manuscript includes an important finding that the expression and function of importin β3 could associate to maintain the dengue viral replication and make the differences between acute and persistent infection. However, there are several concern points in this manuscript as below.
- The authors should provide critical data or evidence that importin β3 can transport NS5 into the nucleus or associate with the nuclear translocation. It has been known that NS5 is transported by importin β1 (KPNA1) in conjugated with importin α as refers in the manuscript [refs 37, 38], and importins β1 and β3 are completely different. This is supported by the evidence that several inhibitors for the importin α/β1 pathway can suppress the nuclear localization of NS5 [refs 45-50]. Also, I could not recognize correlation between the expression levels of importin β3 and the NS5 nuclear localization (Figure 6). Therefore, I would suggest authors to show the relationship between importin β3 and NS5 by any critical references or data. If not, the description (for example lines 335-340 and 505-506) could be misleading in that importin β3 coordinates with the NS5 nuclear localization.
- In relation to the above comments, I am concerned to describe "importin β" instead of "importin β3" in the manuscript. To avoid confusion for readers, authors should clearly describe “importin β3” through the manuscript including the title and abstract, rather than “importin β” (line 326).
- I wonder if it could be shown cell viability at each time point in each MOI in Figure 2 along with the viral titer graph or as supplemental data.
- I would suggest showing the number of experiments in the figure legend in Figure 4.
- Line 544: importin β “gene”, not “gen”.
Round 2
Reviewer 2 Report
All comments I requested have been responded to.